# Low VDAC1 Expression Is Associated with an Aggressive Phenotype and Reduced Overall Patient Survival in Cholangiocellular Carcinoma

**DOI:** 10.3390/cells8060539

**Published:** 2019-06-04

**Authors:** René Günther Feichtinger, Daniel Neureiter, Ralf Kemmerling, Johannes Adalbert Mayr, Tobias Kiesslich, Barbara Kofler

**Affiliations:** 1Research Program for Receptor Biochemistry and Tumor Metabolism, Department of Pediatrics, University Hospital Salzburg of the Paracelsus Medical University, 5020 Salzburg, Austria; b.kofler@salk.at; 2Department of Pediatrics, University Hospital Salzburg of the Paracelsus Medical University, 5020 Salzburg, Austria; h.mayr@salk.at; 3Institute of Pathology, University Hospital Salzburg of the Paracelsus Medical University, 5020 Salzburg, Austria; d.neureiter@salk.at (D.N.); ralf.kemmerling@web.de (R.K.); 4Laboratory for Tumor Biology and Experimental Therapies (TREAT), Institute of Physiology and Pathophysiology, Paracelsus Medical University, 5020 Salzburg, Austria; tobias.kiesslich@pmu.ac.at

**Keywords:** cholangiocellular carcinoma, mitochondria, energy metabolism, oxidative phosphorylation

## Abstract

Cancer cells frequently exhibit dysfunctional oxidative phosphorylation (OXPHOS) and a concomitant increase in glycolytic flux. We investigated the expression of OXPHOS complex subunits and mitochondrial mass in 34 human cholangiocellular carcinomas (CCCs) and adjacent normal tissue by using tissue microarrays. In the tumor periphery, all OXPHOS complexes were reduced except complex I. In addition, significantly lower levels of complex IV were found at the tumor center (*p* < 0.0001). Mitochondrial mass, as indicated by VDAC1 expression, was significantly increased in CCCs compared to corresponding normal tissue (*p* < 0.0001). VDAC1 levels were inversely correlated with UICC (Union Internationale Contre le Cancer) cancer stage classification (*p* = 0.0065). Furthermore, significantly lower VDAC1 was present in patients with lymph node involvement (*p* = 0.02). Consistent with this, patients whose carcinomas expressed VDAC1 at low to moderate levels had significantly reduced survival compared to high expressors (*p* < 0.05). Therefore, low mitochondrial mass is associated with more aggressive CCC. These metabolic features are indicative of a Warburg phenotype in CCCs. This metabolic signature has potential therapeutic implications because tumors with low mitochondrial function may be targeted by metabolic therapies such as a high-fat, low-carbohydrate ketogenic diet.

## 1. Introduction

Cholangiocellular carcinomas (CCCs) are rare but aggressive tumors that display features of biliary differentiation. CCCs comprise approximately 3% of gastrointestinal tumors and have an overall incidence of less than two per 100,000 [1]. According to their anatomical location, CCCs are commonly classified as intrahepatic and extrahepatic tumors, the latter entity being further subdivided into perihilar and distal tumors. In Germany, mortality from intrahepatic CCCs more than tripled between 1998 and 2008 [2]. In line with the increased mortality data, the number of reported cases of intrahepatic CCC also increased between 1970 and 2006 [3]. CCC is one of the most fatal cancers: the median overall survival is 20–28 months and five-year survival rates are about 25% [4,5]. In the United States, the number of CCC-related deaths per annum increased dramatically in the past two decades, surpassing 7000 by 2013 [6]. CCC mortality for those aged 25+ increased 36% between 1999 and 2014 [6]. The only curative option for CCC is surgical resection, but most patients develop a recurrence after resection.

Increased glucose metabolism and uptake is a hallmark of aggressive cancer. Accordingly, elevated ^18^Fluorodesoxyglucose (FDG) uptake was present in 92% of CCCs [7]. This phenomenon, also called the Warburg effect, indicates that cancer cells generate energy predominantly via glycolysis even if sufficient oxygen is present. At first, it seems paradoxical that tumors use ‘inefficient’ glycolysis instead of OXPHOS for energy production. However, there are several explanations for why ATP generation is reprogrammed in cancer cells. First, aerobic glycolysis is not as inefficient as is often reported. Although it is correct that the amount of ATP generated per molecule of glucose is low, the rate of glucose metabolism is high in cancer cells. The production of lactate from glucose occurs 10–100 times faster than the complete oxidation in mitochondria, and the level of ATP production is similar [8]. Thus, rewiring energy metabolism toward glycolysis causes increased generation of lactate. Several studies suggested that lactate facilitates metastasis via production of a microenvironment toxic to normal cells and stimulation of tissue lysis [9,10]. Lactate dehydrogenase A (LDHA) catalyzes the inter-conversion of pyruvate and L-lactate with concomitant inter-conversion of nicotinamide adenine dinucleotide (NADH) and NAD^+^ [11]. Secondly, the Warburg effect has been proposed to be an adaptive mechanism to support the biosynthetic requirements of uncontrolled proliferation. Glucose metabolites serve as carbon sources for anabolic processes. The excess carbon is diverted into branching pathways emanating from glycolysis and is used for the generation of cellular building blocks such as nucleotides, lipids, and proteins [12,13,14,15]. Another theory proposes that tumors shut down OXPHOS to reduce the reactive oxygen species (ROS) burden on their own biomolecules, while maintaining a level necessary for cell signaling [14]. The respiratory chain complexes are the main production sites of ROS, although their individual contributions to this process likely differ, with complex I being the most important in this vicious circle [16].

Immunohistochemical (IHC) staining of OXPHOS complexes of homogeneous tissue samples correlates well with enzymatic analysis, as the OXPHOS system is mainly regulated via the protein amount [17,18,19]. In the present study, we used IHC for technical reasons, foremost because it is nearly impossible to obtain sufficient amounts of frozen CCC tissue for functional evaluation of OXPHOS enzymes. In addition, substantial cellular heterogeneity is present within single tumors. Furthermore, a tumor cell content of over 80% is needed to generate reliable functional data on OXPHOS enzyme activity. IHC staining of heterogeneous samples is the method of choice because it reliably reflects the in vivo situation at the cellular level [18,19].

Therefore, the aim of the present study was to characterize the OXPHOS phenotype of CCC by IHC using tissue microarrays.

## 2. Materials and Methods

### 2.1. Ethics

Human tumors were obtained from the Institute of Pathology, University Hospital Salzburg. The study was performed according to the Austrian Gene Technology Act. Experiments were conducted in accordance with the Helsinki Declaration of 1975 (revised 2013) and the guidelines of the local ethics committee, being no clinical drug trial or epidemiological investigation. All patients signed an informed consent document concerning the surgical intervention. Furthermore, the study did not extend to the examination of individual case records. Patient anonymity was ensured at all times. All analyses on human CCC samples were approved by the local ethics committee (415-EP/73/37-2011).

### 2.2. Samples

To evaluate differences in expression between malignant and corresponding normal tissue, we constructed a tissue microarray (TMA) of formalin-fixed, paraffin-embedded (FFPE) tissue blocks from 34 individuals with CCC. Three punches of each individual were analyzed: the tumor center, the tumor periphery, and the adjacent normal tissue. Samples were analyzed by two professional pathologists and the mean values were taken for statistics.

### 2.3. Clinical Parameters

The following clinical parameters were evaluated: age, sex, overall survival from the day of diagnosis, localization (intrahepatic, perihilar, extrahepatic), growth type (mass forming, periductal, intraductal), tumor size, TNM (tumor, node, metastasis) classification, metastasis, UICC (Union Internationale Contre le Cancer), grading, etiology (Appendix A).

### 2.4. Immunohistochemical Staining of OXPHOS Complex Subunits and VDAC1 (Porin) of FFPE Tissues

All primary antibodies (Table 1) were diluted in Dako antibody diluent with background-reducing components (Dako, Glostrup, Denmark). Immunohistochemistry was performed as described previously [20]. For antigen retrieval, the sections were immersed for 45 min in 1 mM EDTA, 0.05% Tween-20, pH 8, at 95 °C. Tissue sections were incubated for 30 min with the above-listed primary antibodies (Table 1).

### 2.5. Statistical Analysis

For comparison of tumors and normal adjacent tissue, a t-test was applied. For multiple comparisons, one-way ANOVA and Bonferroni correction were applied. The Pearson correlation was applied to analyze potential associations between the evaluated parameters. For analysis of survival, Kaplan–Meier curves were used.

## 3. Results

### 3.1. IHC Scoring

Staining intensities were rated using a scoring system ranging from 0 to 3, with 0 indicating no staining, 1 being mild, 2 moderate, and 3 strong staining. Score values were obtained by multiplying the staining intensity by the percentage of positive cells. The percentage of positive cells was analyzed with 10% increments. For 10–20% of positive cells a median value of 15% was used for statistics. Examples for immunohistochemical scorings are shown in Figure 1.

### 3.2. Expression of VDAC1 and Subunits of the Five OXPHOS Complexes

VDAC1 was used as a marker for the mitochondrial mass. It is highly expressed in the outer mitochondrial membrane which is otherwise relatively sparse of proteins. Therefore it represents the gold standard for determination of the mitochondrial amount. Protein complexes of the OXPHOS are localized in the inner mitochondrial membrane where they transport electrons to generate a proton gradient used by the ATP synthase (complex V) to make ATP. Subunits for each of the five OXPHOS complexes were analyzed in CCCs. Complex I (NADH coenzyme Q oxidoreductase) is the largest multisubunit complex of the OXPHOS system with a molecular mass of 970 kDa consisting of 45 subunits [21,22]. NDUFS4 is an iron–sulfur cluster-containing subunit incorporated during a very late stage of complex I assembly essential for complex I function. Complex II (succinate dehydrogenase) is the smallest complex consisting of four subunits and the only complex exclusively encoded by the nuclear DNA. Complex III (coenzyme Q: cytochrome c-oxidoreductase) consists of 22 subunits. Cytochrome b is the only mtDNA-encoded subunit of complex III [23]. Complex IV (cytochrome c oxidase) represents the last complex of the respiratory chain catalyzing the terminal step in reduction O**_2_**. Three complex IV subunits are encoded by mtDNA. Complex V (ATP synthase) uses the protons translocated by the respiratory chain enzymes for production of ATP [23,24]. Complex I, complex III, and complex IV are furthermore organized in even bigger protein complexes, of which the most abundant one is termed respirasome [25].

Significantly higher VDAC1 expression was observed in the tumor center compared to adjacent normal tissue (*p* < 0.0001) (Figure 1A–C, Figure 2A,C and Figure 3A).

VDAC1 levels in the tumor periphery were similar to those in normal tissue and significantly lower than in the tumor center (*p* < 0.001) (Figure 1A–C and Figure 3A). NDUFS4 (subunit of complex I) expression did not differ between normal tissue and the tumor center or the periphery (Figure 2 and Figure 3B). SDHA (subunit of complex II) levels were significantly lower in the tumor periphery than in the tumor center (*p* < 0.05) and in adjacent normal tissue (*p* < 0.001) (Figure 1D–F and Figure 3C). No significant difference was detected between the tumor center and normal tissue (Figure 1D,E). UQCRC2 (subunit of complex III) expression was lower in the periphery compared to the control tissue (*p* < 0.01) (Figure 3D). MT-CO1 (subunit of complex IV) protein levels were significantly reduced in both the center (*p* < 0.0001) and periphery (*p* < 0.001) of the tumors compared to normal tissue (Figure 3E). Finally, ATP5F1A (subunit of complex V/ATP synthase) was significantly diminished in the tumor periphery (*p* < 0.001) and showed a trend to lower levels in the center, compared to controls (Figure 3F). 

Areas with cells negative for one or more OXPHOS subunits were found in both tumors and normal tissue (Figure 2D–F,M–O): 3% (tumor center), 21% (tumor periphery), and 25% (normal tissue) of the cases showed a loss of VDAC1 in more than 30% of the analyzed cells; 43% (tumor center), 60% (tumor periphery), and 35% (normal tissue) of the cases showed a loss of NDUFS4 (Figure 2F); 30% (tumor center), 54% (tumor periphery), and 11% (normal tissue) showed a loss of SDHA; 34% (tumor center), 54% (tumor periphery), and 10% (normal tissue) showed a loss of UQCRC2; 45% (tumor center) (Figure 2O), 41% (tumor periphery), 0% (normal tissue) showed a loss of MT-CO1; and 23% (tumor center), 30% (tumor periphery), and 0% (normal tissue) showed a loss of ATP5F1A. 

### 3.3. Associations between OXPHOS Subunit and VDAC1 Expression and Clinical Outcome

A significant inverse correlation between the percentage of VDAC1-positive cells and MT-CO1-negative cells was detected (*p* = 0.0093). A significant inverse correlation was found between VDAC1 score values and UICC stage (*p* = 0.0065; R = −0.4855) (Table 2).

VDAC1 levels in the tumor center were lower in cases with lymph node involvement (*p* = 0.0201). The tumor periphery showed a similar trend (mean score values N0 = 162 ± 31 versus N1 = 147 ± 46). The same trend was present when metastasis occurred in the tumor center (mean score value for M0 = 229 ± 54; mean score value for M1 = 186 ± 48) (Table 3). No differences were observed in the tumor periphery with respect to M stage. Significantly lower NDUFS4 levels were present in males compared to females (*p* = 0.0454).

We divided the cases into high/moderate and low expressors. High expressors were defined as having staining intensities above 2. Since, in general, the staining intensities were lower for NDUFS4 and MT-CO1, high expressors for these subunits were defined by staining intensities above 1.5. A significant difference (*p* < 0.05) in survival was observed between these groups (high/moderate vs. low expressors) for VDAC1 expression (Figure 4 and Figure 5).

No correlations were found between survival and expression of any of the markers of differentiation (epithelial: CK7, CK19; mesenchymal: vimentin), cell cycle proteins (p16, p27, p53 and Ki67) and OXPHOS subunits. Kaplan–Meier analysis revealed that individuals with tumors with high VDAC1 expression (staining intensity > 2) had significantly reduced overall survival (*p* < 0.05) compared to low expressors (staining intensity ≤ 2). None of the OXPHOS complexes was significantly associated with survival. Score values, extensities (percent positive/negative cells) and the intensities were used for the overall survival analysis. However, only the intensity of VDAC1 staining was associated with survival. Therefore, we suppose that the staining intensity might be an independent prognostic factor for CCCs indicating more an on–off-phenomenon than a gradient mechanism. 

## 4. Discussion

The analysis of the mutational landscape revealed that TP53, KRAS (Kirsten rat sarcoma viral oncogene), IDH1 (Isocitrate dehydrogenase 1), and PTEN (Phosphatidylinositol 3,4,5-trisphosphate 3-phosphatase and dual-specificity protein phosphatase) are the most frequently mutated genes in intrahepatic CCCs [26]. Numerous studies have shown that p53 is involved in the regulation of many reactions in energy metabolism. p53 can be regarded as a master regulator of energy metabolism since it influences glycolysis, gluconeogenesis, the pentose-phosphate pathway, mitochondrial OXPHOS, and glutamine metabolism [27,28]. Mutations in the oncogene KRAS drive metabolic reprogramming through enhanced glucose uptake and regulation of glutamine metabolism [29]. Additionally, PTEN was shown to regulate several aspects of energy metabolism [30]. IDH1 catalyzes the reversible oxidative decarboxylation of isocitrate to yield α-ketoglutarate (α-KG). Under hypoxic conditions, IDH1 catalyzes the reverse reaction of α-KG to isocitrate, which contributes to citrate production via glutaminolysis [31,32]. In the present study, the levels of SDHA, UQCRC2, MT-CO1 and ATP5F1A were significantly reduced in the tumor periphery compared to the control tissue. Thus, genetic alterations of proteins influencing energy metabolism are clearly central to the pathogenesis of CCC.

Large areas of NDUFS4 negative cells were found in tumors and control tissue, suggesting that this is an early event in tumorigenesis. No patient had large negative regions of MT-CO1 or ATP5F1A in normal tissue. In addition, SDHA and UQCRC2 were affected in only 11% and 10% of the cases, respectively. Mitochondrial DNA was reported to be significantly mutated in CCCs [26,33]. This can partially explain the observed OXPHOS defects in our sample cohort. Moreover, the heteroplasmy of mtDNA mutations might explain the heterogeneity of the OXPHOS subunit expression [34]. According to the COSMIC database, potentially pathogenic mutations in nuclear-encoded OXPHOS subunits are very rare events in CCCs. Therefore, it would be interesting to analyze the mutational landscape (TP53, KRAS, IDH1, PTEN) in relation to the expression of OXPHOS subunits. However, it was not possible to perform a detailed genetic analysis because this was a retrospective study using formalin-fixed paraffin-embedded (FFPE) tissue.

As we found differences in the expression levels between the tumor center and periphery, we hypothesize that within CCCs, several modes of energy generation coexist. Tumor cells at the margin might be more dependent on glucose than tumor cells at the center, as indicated by the lower levels of OXPHOS subunits in the latter. This could be attributable to the fact that newly generated tumor cells at the growth front might not be sufficiently supplied with oxygen, because the generation of new blood vessels requires time. However, this hypothesis is controversial because necrosis is often seen in the tumor center, which is at least partially attributed (in the literature) to low oxygen supply.

Mitochondrial mass was increased in the tumor center compared to adjacent normal tissue, as indicated by VDAC1 staining. A significant inverse correlation found between the percentage of VDAC1-positive cells and MT-CO1-negative cells suggests there may be compensatory upregulation of mitochondrial mass. This phenomenon is well described in individuals with mitochondrial disorders, who frequently have increased mitochondrial content, as indicated by citrate synthase activity, VDAC1 levels, and mtDNA copy number [35]. In addition, oncocytic tumors, which are characterized by complex I defects caused by pathogenic mutations in mtDNA-encoded subunits, show a very pronounced increase in mitochondrial mass [19,36]. In addition, mtDNA content and mass increase in the tumorigenic progression of normal endometrial tissue to hyperplastic tissue to cancerous tissue [37]. MtDNA mutations and deletions have been reported in endometrial tumors [38].

Another possibility is that tumor cells at the periphery might express different sets of proteins compared to tumor cells in the center. The intercellular or cell–cell lactate shuttle hypothesis proposes that lactate is generated and exported from one cell and taken up and utilized by another cell. This mechanism has been described for neurons and astrocytes [39].

A significant inverse correlation was found between VDAC1 expression and UICC tumor stage. This agrees with findings in the literature, and may be explained as follows: aggressive tumors that are highly dependent on glucose should potentially exhibit low mitochondrial mass. In support, we found that VDAC1 levels were lower in cases with lymph node involvement. The same trend to lower VDAC1 levels was also present if metastasis occurred. Significantly lower survival was observed for low/moderate VDAC1 expressors compared to high expressors. That is, lower mitochondrial mass was associated with shorter survival.

The clear association between energy metabolism and CCC development has important therapeutic implications. CCCs might be susceptible to metabolic therapies such as the ketogenic (high-fat, low-carbohydrate) diet, which was recently shown to significantly inhibit tumor growth in numerous xenograft models and patients [40,41,42,43].

## Figures and Tables

**Figure 1 cells-08-00539-f001:**
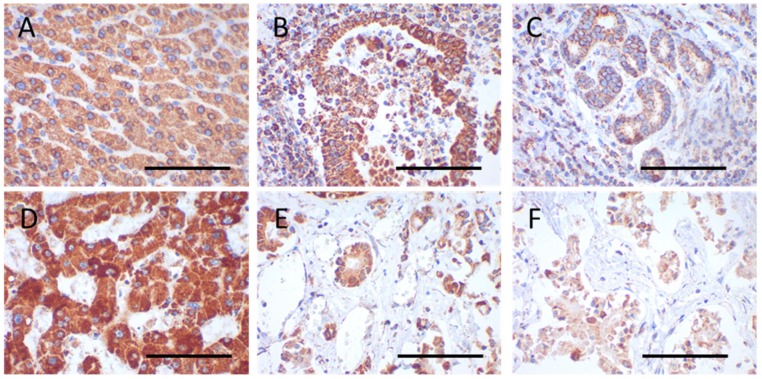
Immunohistochemical staining of VDAC1 and SDHA in the tumor center and periphery and normal adjacent tissue. (**A**–**C**) VDAC1 (case 28); (**D**–**F**) SDHA (case 4); (**A**,**D**) normal tissue; (**B**,**E**) tumor center; (**C**,**F**) tumor periphery. The following score values were evaluated for the images: Score value = intensity × percent positive cells; (**A**) 150 = 2 × 75; (**B**) 295 = 3 × 85; (**C**) 97.5 = 1.5 × 65; (**D**) 255 = 3 × 85; (**E**) 187.5 = 2.5 × 75; (**F**) 130 = 2 × 65. Magnification = 400×. Scale bar = 100 µm.

**Figure 2 cells-08-00539-f002:**
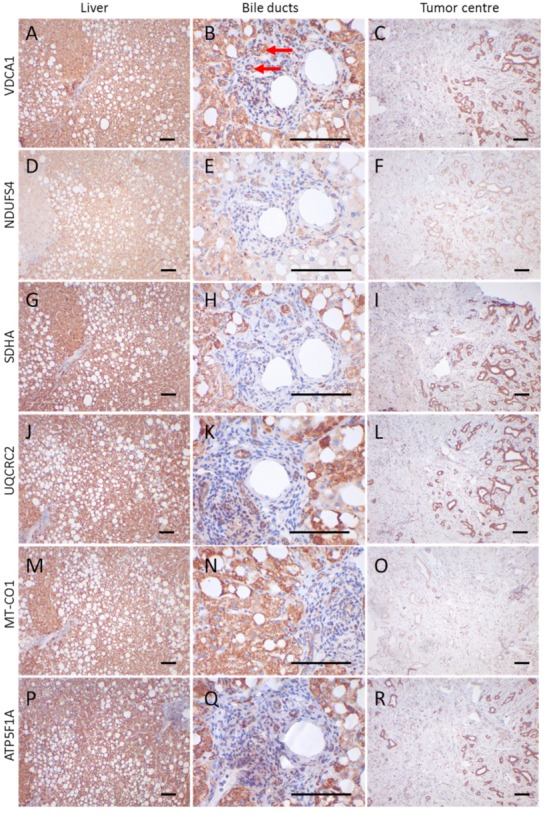
Immunohistochemical staining of a CCC and adjacent control tissue. (**A**,**D**,**G**,**J**,**M**,**P**) liver control tissue; (**B**,**E**,**H**,**K**,**N**,**O**) control bile duct tumor; (**C**,**F**,**I**,**L**,**O**,**R**) tumor center; (**A**–**C**) VDAC1; (**D**–**F**) NDUFS4; (**G**–**I**) SDHA; (**J**–**L**) UQCRC2; (**M**–**O**) MT-CO1; (**P**–**R**) ATP5F1A. Images of the liver and tumor center were taken at 100× magnification. For bile ducts a 400× magnification is shown. Scale bars = 100 µm. Red arrows highlight bile ducts. Case 11, a 59-year-old man, is shown.

**Figure 3 cells-08-00539-f003:**
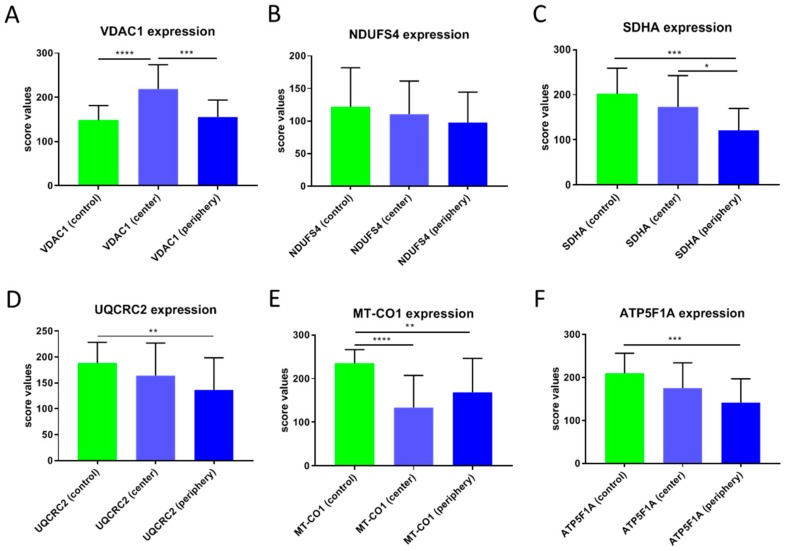
Score values of staining of the OXPHOS complexes and VDAC1 in CCCs. (**A**) VDAC1, (**B**) NDUFS4, (**C**) SDHA, (**D**) UQCRC2, (**E**) MT-CO1, (**F**) ATP5F1A. The mean score values ± SD of the staining of control tissues, tumor center, tumor periphery, and the average of center and periphery are given. **** *p* < 0.0001, *** *p* < 0.001, ** *p* < 0.01, * *p* < 0.05.

**Figure 4 cells-08-00539-f004:**
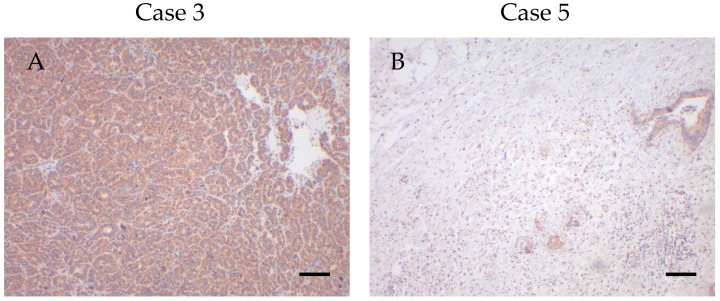
High and low VDAC1 expressors with low and high survival times. (**A**) High VDAC1 expression in case 3 with a survival time of 0.49 months. (**B**) Low VDAC1 expression in case 5 with a survival time of 30.05 months. Magnification 100×. Scale bar = 100 µm.

**Figure 5 cells-08-00539-f005:**
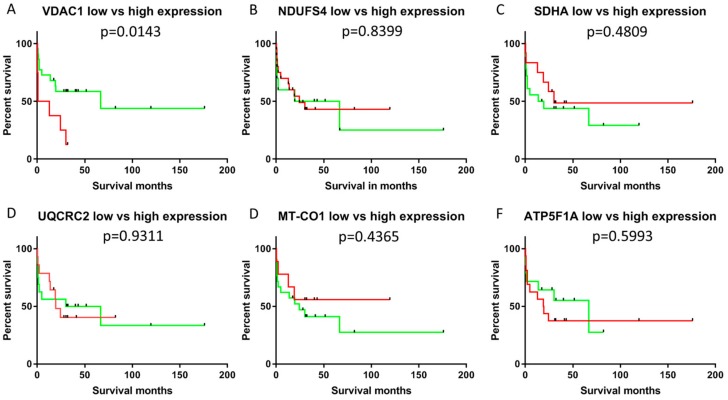
Kaplan–Meier plot of patients with cholangiocellular carcinomas exhibiting low/moderate or high VDAC1 staining intensity. High expressors are shown in red and low expressors in green. (**A**) VDAC1; (**B**) NDUFS4; (**C**) SDHA; (**D**) UQCRC2; (**E**) MT-CO1; (**F**) ATP5F1A. Significantly shorter survival was present in low/moderate expressors (*p* < 0.05).

**Table 1 cells-08-00539-t001:** Antibodies used for immunohistochemical (IHC) staining.

Target Structure/Antigen Specificity (Species)	Vendor	Catalogue No.	Dilution
VDAC1	Abcam	Ab14734	1:2000
NDUFS4 (Complex I)	Abcam	Ab55540	1:1000
SDHA (Complex II)	Abcam	Ab14715	1:2000
UQCRC2 (Complex III)	Abcam	Ab14745	1:1500
MT-CO1 (Complex IV)	Abcam	Ab14705	1:1000
ATP5F1A (Complex V)	Abcam	Ab14748	1:2000
Ck7	Novocastra Laboratories	NCL-L-CK7-OVTL	1:100
Ck19	DakoCytomation	NCL-CK19	1:100
Vimentin	DakoCytomation	M0725	1:2000
Ki67	DakoCytomation	M7249	1:500
p16	mtm laboratories AG	9511	**^a^**
p27	DakoCytomation	M7203	1:100
p53	DakoCytomation	M7001	1:200

^a^ according to the manufacturer’s instructions.

**Table 2 cells-08-00539-t002:** Correlations of OXPHOS subunits and VDAC1 with respect to clinical parameters.

	VDAC1		NDUFS4		SDHA		UQCRC2		MT-CO1		ATP5F1A	
	*p* Value	R Value	*p* Value	R Value	*p* Value	R Value	*p* Value	R Value	*p* Value	R Value	*p* Value	R Value
Age at Diagnosis	0.4683	0.1376	0.6647	−0.0825	0.8635	0.0328	0.2555	−0.2182	0.2166	−0.2366	0.8365	−0.0394
Tumor Size	0.5292	0.1196	0.2929	0.1986	0.3501	0.1768	0.0833	0.3271	0.5767	0.1081	0.6339	−0.0906
TMN Stage	0.1458	−0.2721	0.5047	−0.1267	0.5106	−0.125	0.9133	0.0211	0.8660	0.0328	0.9418	−0.0139
UICC	0.0065	−0.4855	0.3039	−0.1941	0.3659	−0.1711	0.9702	−0.0073	0.5931	−0.1035	0.1066	−0.3005

**Table 3 cells-08-00539-t003:** Mean staining scores of OXPHOS subunits and VDAC1 with respect to clinical parameters.

		VDAC1	*p* Value	NDUFS4	*p* Value	SDHA	UQCRC2	MT-CO1	ATP5F1A
Female	n = 9	215 ± 61		135 ± 51	0.0454	152 ± 84	154 ± 66	141 ± 84	179 ± 57
Male	n = 21	221 ± 54		103 ± 47		185 ± 61	166 ± 65	130 ± 71	175 ± 62
Intrahepatic	n = 16	224 ± 48		122 ± 50		182 ± 66	174 ± 61	148 ± 79	178 ± 54
Perihilar	n = 11	209 ± 64		104 ± 33		154 ± 77	141 ± 68	114 ± 74	170 ± 75
Extrahepatic	n = 3	224 ± 68		93 ± 103		221 ± 14	187 ± 68	123 ± 31	192 ± 36
Mass forming	n = 15	228 ± 49		116 ± 48		180 ± 61	178 ± 62	140 ± 81	172 ± 45
Periductal	n = 15	210 ± 61		109 ± 53		171 ± 77	148 ± 65	127 ± 67	181 ± 73
N stage 0	n = 18	240 ± 44	0.0201	115 ± 61		190 ± 63	177 ± 63	152 ± 76	187 ± 46
N stage 1	n = 12	187 ± 57		108 ± 29		153 ± 73	139 ± 62	104 ± 62	160 ± 75
M stage 0	n = 23	229 ± 54		116 ± 51		176 ± 71	161 ± 67	130 ± 82	175 ± 64
M stage 1	n = 7	186 ± 48		99 ± 45		174 ± 67	167 ± 59	147 ± 27	180 ± 49
Grade 2	n = 17	205 ± 63		105 ± 45		162 ± 62	150 ± 66	108 ± 67	160 ± 56
Grade 3	n = 11	239 ± 39		117 ± 53		179 ± 74	171 ± 65	164 ± 78	198 ± 51
R-status 0	n = 19	226 ± 52		106 ± 42		182 ± 62	176 ± 54	131 ± 60	167 ± 51
R-status 1	n = 11	207 ± 60		124 ± 62		164 ± 82	140 ± 76	140 ± 98	192 ± 73

The non-parametric Mann–Whitney test was used for analysis.

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
