# Peer review of "Low VDAC1 Expression Is Associated with an Aggressive Phenotype and Reduced Overall Patient Survival in Cholangiocellular Carcinoma"

_cells, 2019, doi:10.3390/cells8060539_

Round 1

Reviewer 1 Report

The study entitled “Low VDAC1 expression is associated with an aggressive phenotype and reduced overall patient survival in cholangiocellular carcinoma” shows that human cholangiocellular carcinomas expressing VDAC1 at low to moderate levels had significantly reduced survival compared to high expressors. The subject of the manuscript is interesting and the study appears well performed and the manuscript is well-written and organized. Personally, I think that the manuscript is suitable for publication and that Cells would be an appropriate place for it to be published.

I would like to suggest the authors to address the following points to improve the manuscript.

Minor Revision

- I suggest to remove in the materials and methods, section 2.5, the following description: The specificity of the antibodies used was previously shown by Western blot analysis in multiple publications: ([21]; Figure 3), ([22], Figure 2); NDUFS4 ([21]; Figure3), ([22], Figure 2); SDHA ([21]; Figure 3), ([22], Figure 2); UQCRC2 ([21]; Figure 3), ([22], Figure 2); MT-CO1 ([23], Figure 2); ATP5F1A ([21]; Figure 3), ([24]; Figure 3).

- Figure 1 should be inserted in the results as preliminary.

- In the result section a better description of the five OXPHOS complexes should be added before discussing the results obtained for each complex.

Author Response

We would like to thank the reviewers for their constructive and fruitful suggestions. We also added a word Version of the reviewer reply for download.

Reviewer 1-Point 1: I suggest to remove in the materials and methods, section 2.5, the following description: The specificity of the antibodies used was previously shown by Western blot analysis in multiple publications: ([21]; Figure 3), ([22], Figure 2); NDUFS4 ([21]; Figure3), ([22], Figure 2); SDHA ([21]; Figure 3), ([22], Figure 2); UQCRC2 ([21]; Figure 3), ([22], Figure 2); MT-CO1 ([23], Figure 2); ATP5F1A ([21]; Figure 3), ([24]; Figure 3).

Reply Reviewer 1-Point 1: We removed the mentioned part from section 2.5 (see manuscript with tracked changes)

Reviewer 1-Point 2: Figure 1 should be inserted in the results as preliminary.

Reply Reviewer 1-Point 2: We moved Figure 1 to the Results section and added the headline “IHC scoring”.

Reviewer 1-Point 3: In the results section a better description of the five OXPHOS complexes should be added before discussing the results obtained for each complex.

Reply Reviewer 1-Point 3: According to reviewers suggestion we added a short description of the OXPHOS complexes before presentation of the results and added some references excellently explaining the OXPHOS organization in detail (see manuscript with tracked changes).

VDAC1 was used a marker for the mitochondrial mass. It is highly expressed in the outer mitochondrial membrane which is otherwise relatively sparse of proteins. Therefore it represents the gold standard for determination of the mitochondrial amount. Protein complexes of the OXPHOS are localized in the inner mitochondrial membrane where they transport electrons to generate a proton gradient used by the ATP synthase (complex V) to make ATP. Subunits for each of the five OXPHOS complexes were analyzed in CCCs. Complex I (NADH coenzyme Q oxidoreductase) is the largest multisubunit complex of the OXPHOS system with a molecular mass of 970 kDa consisting by 45 subunits [25,26]. NDUFS4 is an iron-sulfur cluster containing subunit incorporated during a very late stage of complex I assembly essential for complex I function. Complex II (succinate dehydrogenase) is the smallest complex consisting of 4 subunits and the only complex exclusively encoded by the nuclear DNA. Complex III  (coenzyme Q : cytochrome c-oxidoreductase) is consisting of 22 subunits. Cytochrome b is the only mtDNA encoded subunit of complex III [27]. Complex IV (cytochrome c oxidase) represents the last complex of respiratory chain catalyzing the terminal step in reduction O2. Three complex IV subunits are encoded by mtDNA. Complex V (ATP synthase) uses the protons translocated by the respratory chain enzymes for production of ATP [27,28]. Complex I, complex III and complex IV are furthermore organized in even bigger protein complexes the most abundant one termed respirasome [29].

Reviewer 2-Point 1.1: Two minor points: the running title promises too much; OXPHOS is only indirectly implied, not examined.

Reply Reviewer 2-Point 1.1: According to reviewer we changed the running title to “Subunits of the oxidative phosphorylation in cholangiocellular carcinoma”.

Reviewer 2-Point 1.2: Line 100: was overall survival calculated from day of diagnosis? The reason I ask is that Fig 3 indicates a patient with 0.49 month survival = 14.7 days.

Reply Reviewer 2-Point 1.1:  Yes, the overall survival was calculated from the day of diagnosis. We clarified this point in the manuscript. The first sentence of chapter 2.3 was changed to: “The following clinical parameters were evaluated: age, sex, overall survival from the day of diagnosis,……”

Reviewer 2-Point 2: Figs 2-3: How was percentage positive cells calculated? Did you analyze the data with 5 or 10 percentage point increments? Can you be sure it is possible to distinguish between, for instance, 65% and 70%?

Reply Reviewer 2-Point 2: Data were analyzed with 10% increments. For example 50-60% positive or 80-90% positive. The respective median (55 or 85%) was used for analysis. A difference of 10% can be distinguished. A difference of 5% might not be visible.

We also included the following sentence in the methods section. “Samples were analyzed by two professional pathologists and the mean values were taken for statistics.”

The following sentence was included in the results section to clarify this point: “The percentage of positive cells was analyzed with 10% increments. For 10-20% positive cells a median value of 15% was used for statistics.”

Reviewer 2-Point 3. Since VDAC has many functions in cell death and survival, metabolism etc, there is some basis for questioning whether there is a direct correlation between VDAC and mitochondrial content. You show a statistical inverse correlation between % VDAC-positivity and MT-CO1-negativity. Were areas (areas, not samples) that were VDAC-negative in actual fact also negative for OXPHOS subunits?

Reply Reviewer 2-Point 3: VDAC1 represents the gold standard as a mitochondrial loading control for western blotting, IF and FFPE IHC. There are already hundreds of articles about mitochondrial diseases using VDAC1 as an indicator of the mitochondrial mass. The reason is that the mitochondrial outer membrane is relatively sparse of proteins with VDAC1 one of the only highly expressed proteins. Another option is citrate synthase in the mitochondrial matrix where good antibodies are available. The fact that VDAC1 is a very good indicator for the mitochondrial mass is also underlined by its excellent correlation with citrate synthase activity. Some researchers think that they can use PGC-1alpha or TFAM as loading controls. However, this makes no sense in our opinion since PGC-1alpha is not localized in mitochondria. For example DGUOK patients with low TFAM levels caused by mtDNA depletion have a normal mitochondrial mass as indicated by citrate synthase activity.

We added a short sentence why VDAC1 is usually used as a loading control.” VDAC1 was used a marker for the mitochondrial mass. It is highly expressed in the outer mitochondrial membrane which is otherwise relatively sparse of proteins. Therefore it represents the gold standard for determination of the mitochondrial amount.”

The reviewer is totally right that VDAC1 negative areas were also negative for the OXPHOS subunits. If no mitochondria are present as indicated by VDAC1 also the OXPHOS subunits are missing. However, these areas were usually very small. The median percentage of VDAC1 positive tumor cells was 87%.

Reviewer 2-Point 4. This correlation was based on positivity only, not intensity or score. In cases with low VDAC positivity and high intensity, were these areas also MT-CO1 negative?  Or positive?

Reply Reviewer 2-Point 4: The median percentage of VDAC1 positive tumor cells was 87%. Only one case (case 31) had a more severe loss of VDAC1 with a positivity of 65%. In this case a COX- deficiency was found in 87,5% of the cells. Although the score value was extremely low for all cells (intensity 1; positivity 12,5%). Areas with no VDAC1 show also a loss of the other complexes. Areas with a high VDAC1 expression can show a loss of COX. Usually it depends on the degree of COX down regulation. We suppose that a certain threshold might exist. If the loss of COX or another complex is severe a compensatory program is activated and the mitochondrial mass increases.

 Reviewer 2-Point 5. Lines 170-71: does "VDAC expression" signify positivity or the score value?

Reply Reviewer 2-Point 5: According to the reviewer comment we clarified this point and changed the sentence to: “A significant inverse correlation was found between VDAC1 expression score values and UICC stage (p=0.0065; R=-0.4855) (Table 2).”

Reviewer 2-Point 6. Lines 174-75: here, it is clearly said that it is based on staining scores. But which part of the tumor? The central sample? Combination of central and periphery? Would this affect the conclusions/hypotheses?

Reply Reviewer 2-Point 6: According to the suggestion of the reviewer this point was clarified and the sentence changed “VDAC1 levels in the tumor center were lower in cases with lymph node involvement  (p=0.0201).”

According to reviewer we also calculated the values for the tumor periphery. The trend to lower levels of VDAC1 is also present in the tumor periphery. Mean score values N0= 162+/-31 versus N1=147+/-46. The mean score values for M0 vs M1 did not differ in the tumor periphery. 

The text was modified according to the reviewer suggestion: “VDAC1 levels in the tumor center were lower in cases with lymph node involvement (p=0.0201). The tumor periphery showed a similar trend (mean score values N0=162±31 versus N1=147±46). The same trend was present when metastasis occurred in the tumor center (mean score value for M0 = 229±54; mean score value for M1 = 186±48) (Table 3). No differences (mean 155 versus 156) were present in the tumor periphery with respect to M stage.”

Reviewer 2-Point 7. Lines 180-84 and Fig.5: here, the data are based on staining intensities, not scores. Why? Does this not skew the results?

We included the score values, extensities (percent positive/negative cells) and the intensities in the overall survival analysis. However, only the intensity of VDAC1 was associated with survival. Therefore, we suppose that the staining intensity might be an independent prognostic factor for CCCs indicating more an on-off-phenomenon than a gradient mechanism. A bias can be excluded because all three parameters were analyzed. Since a low score value can be either explained by a low intensity or positivity or both we also analyzed the two parameters.

The following statement was included in the results section: “Score values, extensities (percent positive/negative cells) and the intensities were used for the overall survival analysis. However, only the intensity of VDAC1 was associated with survival. Therefore, we suppose that the staining intensity might be an independent prognostic factor for CCCs indicating more an on-off-phenomenon than a gradient mechanism.”

Reviewer 2-Point 8: Regarding negative stainings in control tissue, can you for comparison provide information from databases on expression of these proteins in normal tissue? Cholangiocytes would for instance appear to require high ATP production.

Reply Reviewer 2-Point 8: We agree with the reviewer that according to our results cholangiocytes should require a high energy production. However, there are no protein expression data regarding this special cell type. Data regarding the expression of the proteins in normal tissues can be found in proteinatlas. The liver stainings shown in the proteinatlas sometimes include bile ducts. These bile ducts show a high expression of the OXPHOS subunits consistent with our results for example NDUFS4.

https://www.proteinatlas.org/ENSG00000164258-NDUFS4/tissue

https://www.proteinatlas.org/ENSG00000073578-SDHA/tissue

https://www.proteinatlas.org/ENSG00000140740-UQCRC2/tissue

https://www.proteinatlas.org/ENSG00000198804-MT-CO1/tissue

https://www.proteinatlas.org/ENSG00000152234-ATP5A1/tissue

https://www.proteinatlas.org/search/vdac1

RNA expression studies are not very useful because cell mixtures are usually analyzed.

Reviewer 2-Point 9. Line 257: another minor point, but in view of the clearly great range of metabolic phenotypes in and around the tumor, would it not be difficult to target this disease through diet?

Reply Reviewer 2-Point 9: It should be always kept in mind that the ketogenic diet should be used as an additional treatment strategy to conventional therapies. It is a save therapy without major side effects. In addition, feeding tumors with glucose infusions what is done since decades might be the worst option albeit it might be needed in some situations during tumor therapy. We agree with the reviewer that there might be metabolic flexibility however the effect of the ketogenic diet is also based on other mechanism. For example it can lower insulin and IGF-1, which are known growth factors for tumors.

Reviewer 2 Report

This study reports expression levels of selected mitochondrial proteins and their association with location and survival in cholangiocellular carcinoma (CCC). It is of interest for understanding metabolic and mitochondrial alterations in tumor progression, both in CCC and for comparison with other cancers. On the whole, the experiments and data are relevant and the paper well written. However, for the reader to fully understand what has been done and the subsequent interpretations, the following points should be clarified.

1. Two minor points: the running title promises too much; OXPHOS is only indirectly implied, not examined. Line 100: was overall survival calculated from day of diagnosis? The reason I ask is that Fig 3 indicates a patient with 0.49 month survival = 14.7 days.

2. Figs 2-3: How was percentage positive cells calculated? Did you analyze the data with 5 or 10 percentage point increments? Can you be sure it is possible to distinguish between, for instance, 65% and 70%? 

3. Since VDAC has many functions in cell death and survival, metabolism etc, there is some basis for questioning whether there is a direct correlation between VDAC and mitochondrial content. You show a statistical inverse correlation between % VDAC-positivity and MT-CO1-negativity. Were areas (areas, not samples) that were VDAC-negative in actual fact also negative for OXPHOS subunits?

4. This correlation was based on positivity only, not intensity or score. In cases with low VDAC positivity and high intensity, were these areas also MT-CO1 negative?  Or positive?

5. Lines 170-71: does "VDAC expression" signify positivity or the score value?

6. Lines 174-75: here, it is clearly said that it is based on staining scores. But which part of the tumor? The central sample? Combination of central and periphery? Would this affect the conclusions/hypotheses?

7. Lines 180-84 and Fig.5: here, the data are based on staining intensities, not scores. Why? Does this not skew the results?

8. Regarding negative stainings in control tissue, can you for comparison provide information from databases on expression of these proteins in normal tissue? Cholangiocytes would for instance appear to require high ATP production.

9. Line 257: another minor point, but in view of the clearly great range of metabolic phenotypes in and around the tumor, would it not be difficult to target this disease through diet?

Author Response

We would like to thank the reviewers for their constructive and fruitful suggestions. We also added a word version of reviewer reply for download.

Reviewer 1-Point 1: I suggest to remove in the materials and methods, section 2.5, the following description: The specificity of the antibodies used was previously shown by Western blot analysis in multiple publications: ([21]; Figure 3), ([22], Figure 2); NDUFS4 ([21]; Figure3), ([22], Figure 2); SDHA ([21]; Figure 3), ([22], Figure 2); UQCRC2 ([21]; Figure 3), ([22], Figure 2); MT-CO1 ([23], Figure 2); ATP5F1A ([21]; Figure 3), ([24]; Figure 3).

Reply Reviewer 1-Point 1: We removed the mentioned part from section 2.5 (see manuscript with tracked changes)

Reviewer 1-Point 2: Figure 1 should be inserted in the results as preliminary.

Reply Reviewer 1-Point 2: We moved Figure 1 to the Results section and added the headline “IHC scoring”.

Reviewer 1-Point 3: In the results section a better description of the five OXPHOS complexes should be added before discussing the results obtained for each complex.

Reply Reviewer 1-Point 3: According to reviewers suggestion we added a short description of the OXPHOS complexes before presentation of the results and added some references excellently explaining the OXPHOS organization in detail (see manuscript with tracked changes).

VDAC1 was used a marker for the mitochondrial mass. It is highly expressed in the outer mitochondrial membrane which is otherwise relatively sparse of proteins. Therefore it represents the gold standard for determination of the mitochondrial amount. Protein complexes of the OXPHOS are localized in the inner mitochondrial membrane where they transport electrons to generate a proton gradient used by the ATP synthase (complex V) to make ATP. Subunits for each of the five OXPHOS complexes were analyzed in CCCs. Complex I (NADH coenzyme Q oxidoreductase) is the largest multisubunit complex of the OXPHOS system with a molecular mass of 970 kDa consisting by 45 subunits [25,26]. NDUFS4 is an iron-sulfur cluster containing subunit incorporated during a very late stage of complex I assembly essential for complex I function. Complex II (succinate dehydrogenase) is the smallest complex consisting of 4 subunits and the only complex exclusively encoded by the nuclear DNA. Complex III  (coenzyme Q : cytochrome c-oxidoreductase) is consisting of 22 subunits. Cytochrome b is the only mtDNA encoded subunit of complex III [27]. Complex IV (cytochrome c oxidase) represents the last complex of respiratory chain catalyzing the terminal step in reduction O2. Three complex IV subunits are encoded by mtDNA. Complex V (ATP synthase) uses the protons translocated by the respratory chain enzymes for production of ATP [27,28]. Complex I, complex III and complex IV are furthermore organized in even bigger protein complexes the most abundant one termed respirasome [29].

Reviewer 2-Point 1.1: Two minor points: the running title promises too much; OXPHOS is only indirectly implied, not examined.

Reply Reviewer 2-Point 1.1: According to reviewer we changed the running title to “Subunits of the oxidative phosphorylation in cholangiocellular carcinoma”.

Reviewer 2-Point 1.2: Line 100: was overall survival calculated from day of diagnosis? The reason I ask is that Fig 3 indicates a patient with 0.49 month survival = 14.7 days.

Reply Reviewer 2-Point 1.1:  Yes, the overall survival was calculated from the day of diagnosis. We clarified this point in the manuscript. The first sentence of chapter 2.3 was changed to: “The following clinical parameters were evaluated: age, sex, overall survival from the day of diagnosis,……”

Reviewer 2-Point 2: Figs 2-3: How was percentage positive cells calculated? Did you analyze the data with 5 or 10 percentage point increments? Can you be sure it is possible to distinguish between, for instance, 65% and 70%?

Reply Reviewer 2-Point 2: Data were analyzed with 10% increments. For example 50-60% positive or 80-90% positive. The respective median (55 or 85%) was used for analysis. A difference of 10% can be distinguished. A difference of 5% might not be visible.

We also included the following sentence in the methods section. “Samples were analyzed by two professional pathologists and the mean values were taken for statistics.”

The following sentence was included in the results section to clarify this point: “The percentage of positive cells was analyzed with 10% increments. For 10-20% positive cells a median value of 15% was used for statistics.”

Reviewer 2-Point 3. Since VDAC has many functions in cell death and survival, metabolism etc, there is some basis for questioning whether there is a direct correlation between VDAC and mitochondrial content. You show a statistical inverse correlation between % VDAC-positivity and MT-CO1-negativity. Were areas (areas, not samples) that were VDAC-negative in actual fact also negative for OXPHOS subunits?

Reply Reviewer 2-Point 3: VDAC1 represents the gold standard as a mitochondrial loading control for western blotting, IF and FFPE IHC. There are already hundreds of articles about mitochondrial diseases using VDAC1 as an indicator of the mitochondrial mass. The reason is that the mitochondrial outer membrane is relatively sparse of proteins with VDAC1 one of the only highly expressed proteins. Another option is citrate synthase in the mitochondrial matrix where good antibodies are available. The fact that VDAC1 is a very good indicator for the mitochondrial mass is also underlined by its excellent correlation with citrate synthase activity. Some researchers think that they can use PGC-1alpha or TFAM as loading controls. However, this makes no sense in our opinion since PGC-1alpha is not localized in mitochondria. For example DGUOK patients with low TFAM levels caused by mtDNA depletion have a normal mitochondrial mass as indicated by citrate synthase activity.

We added a short sentence why VDAC1 is usually used as a loading control.” VDAC1 was used a marker for the mitochondrial mass. It is highly expressed in the outer mitochondrial membrane which is otherwise relatively sparse of proteins. Therefore it represents the gold standard for determination of the mitochondrial amount.”

The reviewer is totally right that VDAC1 negative areas were also negative for the OXPHOS subunits. If no mitochondria are present as indicated by VDAC1 also the OXPHOS subunits are missing. However, these areas were usually very small. The median percentage of VDAC1 positive tumor cells was 87%.

Reviewer 2-Point 4. This correlation was based on positivity only, not intensity or score. In cases with low VDAC positivity and high intensity, were these areas also MT-CO1 negative?  Or positive?

Reply Reviewer 2-Point 4: The median percentage of VDAC1 positive tumor cells was 87%. Only one case (case 31) had a more severe loss of VDAC1 with a positivity of 65%. In this case a COX- deficiency was found in 87,5% of the cells. Although the score value was extremely low for all cells (intensity 1; positivity 12,5%). Areas with no VDAC1 show also a loss of the other complexes. Areas with a high VDAC1 expression can show a loss of COX. Usually it depends on the degree of COX down regulation. We suppose that a certain threshold might exist. If the loss of COX or another complex is severe a compensatory program is activated and the mitochondrial mass increases.

 Reviewer 2-Point 5. Lines 170-71: does "VDAC expression" signify positivity or the score value?

Reply Reviewer 2-Point 5: According to the reviewer comment we clarified this point and changed the sentence to: “A significant inverse correlation was found between VDAC1 expression score values and UICC stage (p=0.0065; R=-0.4855) (Table 2).”

Reviewer 2-Point 6. Lines 174-75: here, it is clearly said that it is based on staining scores. But which part of the tumor? The central sample? Combination of central and periphery? Would this affect the conclusions/hypotheses?

Reply Reviewer 2-Point 6: According to the suggestion of the reviewer this point was clarified and the sentence changed “VDAC1 levels in the tumor center were lower in cases with lymph node involvement  (p=0.0201).”

According to reviewer we also calculated the values for the tumor periphery. The trend to lower levels of VDAC1 is also present in the tumor periphery. Mean score values N0= 162+/-31 versus N1=147+/-46. The mean score values for M0 vs M1 did not differ in the tumor periphery. 

The text was modified according to the reviewer suggestion: “VDAC1 levels in the tumor center were lower in cases with lymph node involvement (p=0.0201). The tumor periphery showed a similar trend (mean score values N0=162±31 versus N1=147±46). The same trend was present when metastasis occurred in the tumor center (mean score value for M0 = 229±54; mean score value for M1 = 186±48) (Table 3). No differences (mean 155 versus 156) were present in the tumor periphery with respect to M stage.”

Reviewer 2-Point 7. Lines 180-84 and Fig.5: here, the data are based on staining intensities, not scores. Why? Does this not skew the results?

We included the score values, extensities (percent positive/negative cells) and the intensities in the overall survival analysis. However, only the intensity of VDAC1 was associated with survival. Therefore, we suppose that the staining intensity might be an independent prognostic factor for CCCs indicating more an on-off-phenomenon than a gradient mechanism. A bias can be excluded because all three parameters were analyzed. Since a low score value can be either explained by a low intensity or positivity or both we also analyzed the two parameters.

The following statement was included in the results section: “Score values, extensities (percent positive/negative cells) and the intensities were used for the overall survival analysis. However, only the intensity of VDAC1 was associated with survival. Therefore, we suppose that the staining intensity might be an independent prognostic factor for CCCs indicating more an on-off-phenomenon than a gradient mechanism.”

Reviewer 2-Point 8: Regarding negative stainings in control tissue, can you for comparison provide information from databases on expression of these proteins in normal tissue? Cholangiocytes would for instance appear to require high ATP production.

Reply Reviewer 2-Point 8: We agree with the reviewer that according to our results cholangiocytes should require a high energy production. However, there are no protein expression data regarding this special cell type. Data regarding the expression of the proteins in normal tissues can be found in proteinatlas. The liver stainings shown in the proteinatlas sometimes include bile ducts. These bile ducts show a high expression of the OXPHOS subunits consistent with our results for example NDUFS4.

https://www.proteinatlas.org/ENSG00000164258-NDUFS4/tissue

https://www.proteinatlas.org/ENSG00000073578-SDHA/tissue

https://www.proteinatlas.org/ENSG00000140740-UQCRC2/tissue

https://www.proteinatlas.org/ENSG00000198804-MT-CO1/tissue

https://www.proteinatlas.org/ENSG00000152234-ATP5A1/tissue

https://www.proteinatlas.org/search/vdac1

RNA expression studies are not very useful because cell mixtures are usually analyzed.

Reviewer 2-Point 9. Line 257: another minor point, but in view of the clearly great range of metabolic phenotypes in and around the tumor, would it not be difficult to target this disease through diet?

Reply Reviewer 2-Point 9: It should be always kept in mind that the ketogenic diet should be used as an additional treatment strategy to conventional therapies. It is a save therapy without major side effects. In addition, feeding tumors with glucose infusions what is done since decades might be the worst option albeit it might be needed in some situations during tumor therapy. We agree with the reviewer that there might be metabolic flexibility however the effect of the ketogenic diet is also based on other mechanism. For example it can lower insulin and IGF-1, which are known growth factors for tumors.